# Designing an Effective Front-of-Package Warning Label for Food and Drinks High in Added Sugar, Sodium, or Saturated Fat in Colombia: An Online Experiment

**DOI:** 10.3390/nu12103124

**Published:** 2020-10-13

**Authors:** Lindsey Smith Taillie, Marissa G. Hall, Luis Fernando Gómez, Isabella Higgins, Maxime Bercholz, Nandita Murukutla, Mercedes Mora-Plazas

**Affiliations:** 1Carolina Population Center and Department of Nutrition, Gillings School of Global Public Health, University of North Carolina, Chapel Hill, NC 27599, USA; taillie@unc.edu; 2Department of Health Behavior, Gillings School of Global Public Health, Lineberger Comprehensive Cancer Center, Carolina Population Center, University of North Carolina, Chapel Hill, NC 27599, USA; mghall@unc.edu; 3Facultad de Medicina, Pontificia Universidad Javeriana, 110231 Bogotá, Colombia; L.gomezg@javeriana.edu.co; 4Carolina Population Center, University of North Carolina, Chapel Hill, NC 27516, USA; ihiggins@email.unc.edu (I.H.); bercholz@email.unc.edu (M.B.); 5Vital Strategies, New York, NY 27599, USA; NMurukutla@vitalstrategies.org; 6Departamento de Nutrición Humana, Universidad Nacional de Colombia, 11001 Bogotá, Colombia

**Keywords:** front-of-package labeling, sugar-sweetened beverages, added sugar, obesity prevention, food policy, Latin America

## Abstract

Policies to require warnings on the front of food and drinks high in nutrients of concern (e.g., added sugar, sodium, or saturated fat) are becoming increasingly common as an obesity prevention strategy. Colombia, a country with growing prevalence of obesity, is considering implementing a similar policy. The objective of this study was to assess perceptions and reactions to different warning designs. We conducted a randomized experiment in an online panel of adults age > 18y (*n* = 1997). Participants were randomized to view one of four labels: a control label (barcode), an octagon warning, a circle warning, and a triangle warning. Participants viewed their randomly assigned label on a series of products and answered questions (continuous outcomes ranged from 1–4). Compared to the control, all warnings led to higher perceived message effectiveness (increase in mean from 1.79 in the control to 2.59–2.65 in the warning conditions, *p* < 0.001), a higher percentage of participants who correctly identified products high in nutrients of concern (from 48% in the control condition to 84–89% in the warning conditions, *p* < 0.001), and reduced intentions to purchases these products (decrease in mean from 2.59 to 1.99–2.01 in the warning conditions, *p* < 0.001). Relative to the control, warnings performed similarly across education levels, suggesting this policy would be equitable in Colombia. Looking at differences by warning type, the pattern of results suggested that the octagon warnings performed best. After viewing all label types, 49% of participants selected the octagon warning as the one that most discouraged them from consuming products high in nutrients of concern, while 21% and 27% selected the circle and triangle warning. Colombian policymakers should consider the octagon warning as part of a front-of-package labeling policy to help consumers identify and reduce consumption of foods and drinks high in nutrients of concern.

## 1. Introduction

Colombia is a country with growing levels of overweight and obesity [1,2] and related non-communicable diseases such as cardiovascular disease, diabetes, and cancer [3,4]. The increase in overweight and obesity has been accompanied by a diet characterized by high levels of sugar-sweetened beverages (SSBs) and other foods high in nutrients of concern (e.g., high in added sugar, added sodium, or added saturated fat) [5,6]. For example, a recent study found that of the Colombian packaged food supply, approximately 66 to 80% of packaged foods and drinks exceed thresholds for nutrients of concern set by international nutritional profile models [7]. This trend towards SSBs and other foods high in added sugar, sodium, or saturated fat is concerning, considering that these foods are more likely to be ultra-processed [8,9,10] and a large and growing body of evidence has linked consumption of these products to poorer dietary quality, increased calorie intake, and excess weight gain [11,12,13].

Front-of-package labels have increasingly been recognized by scholars, advocates, and policymakers as a promising policy option to reduce excess consumption of these products [14]. In particular, warnings may be an especially effective front-of-package labeling option because they are easy for consumers to understand and quickly interpret, help them identify products that are unhealthy, and discourage them from purchasing these products [15]. A recent meta-analysis of experimental studies found that front-of-package warnings reduce hypothetical and actual purchasing and consumption of SSBs [16]. Moreover, this policy is rapidly being implemented across the region: regulations requiring front-of-package warnings on nutrients of concern have been implemented in Chile, Peru, and Mexico, and passed in Uruguay and Brazil. Initial results from Chile (the only country to be evaluated to date) show that policies including warnings have been associated with behavior change [17], including sizeable reductions in SSB purchases [18], in addition to reformulation of products to reduce sugar and sodium content [19].

In 2020, the Colombian Ministry of Health announced its intention to implement a front-of-package labeling system which includes a warning on unhealthy foods [20], and several bills introducing front-of-package warnings have also been presented in Congress. However, it is still unclear which warning design would be most effective in the Colombian population to help consumers identify unhealthy products and discourage them from selecting such products. For example, the Chilean warning includes octagons with the text “high in [sugar, sodium, saturated fat, or calories].” The Peruvian warning also includes a similar octagon shape and message but adds an additional message to avoid excess levels of consumption. Still other options include the use of icons to signal the nutrient of concern (e.g., a saltshaker to represent sodium). 

In addition, one concern about nutrient labeling is that it would not serve those who may need it the most, such as low-educated populations who may be less likely to use or understand the back-of-package nutrition facts panel information. It is currently unknown whether the current set of proposed warning designs differentially impacts high vs. low-educated Colombians. 

To inform front-of-package labeling policies in Colombia, the objective of this study was to examine consumers’ perceptions of and reactions to different warning designs and determine whether the effect of the warning differs by educational level. 

## 2. Materials and Methods 

The study was approved and determined to be exempt by the Institutional Review Board of the University of North Carolina, Chapel Hill (UNC) on 27 March 2020 (IRB# 20-0401), and it was designated as exempt from review at Javeriana University. The protocol was pre-registered at https://aspredicted.org/qt4bu.pdf. 

### 2.1. Participants

In March and April 2020, we recruited an convenience sample to participate in an online experiment (*n* = 1997). We recruited participants through Offerwise, an online consumer research firm with a panel of more than 300,000 participants in Colombia. Inclusion criteria included currently residing in Colombia and being at least 18 years old. Offerwise used purposive sampling to create a sample similar to the Colombian population with regards to gender, age (18–24, 25–34, 35–44, 45–54, 55+), region (Atlantic, Bogota, Central, Oriental, Orinoquia, and Pacific), and education (High school graduate or less, College degree or higher). Participants earned agreed upon incentives from Offerwise for completing the study.

### 2.2. Warning Development

The final warnings and barcode control are shown in Figure 1.

Because the objective of this research was to inform active policy discussions, the warnings were based on modified versions of warning label systems that had appeared in previous labeling bills presented to Congress or are being considered by advocacy groups and the Colombian Ministry of Health. Warnings were selected with the guidance of a coalition of nutrition scholars and health advocacy organizations in Bogota, Colombia. 

Warnings in the non-control condition targeted three nutrients: Sugar, sodium, and saturated fat, because these are the nutrients targeted by several regional nutritional profile models [21,22]. All warnings included Spanish text to inform consumers that the product contains high levels of sugar, salt, or saturated fat, and contained a disclosure that the label was authorized by the Colombian Ministry of Health [“MINSALUD”]. 

Similar to the existing front-of-package labels in Chile and Peru, one warning consisted of a black octagon with text about high or excessive levels of the nutrient([in this case, we used, “EXCESS SUGARS”), and, similar to the existing Peruvian warning, an additional statement in a rectangle that instructs consumers to avoid high levels of consumption of a product (“AVOID HIGH CONSUMPTION”). The second warning consisted of black triangles contained within a white box. In addition to one triangle containing an exclamation point (to signal warning) [23], the box contained additional triangles(s) with text about excessive levels of a nutrient (EXCESS SUGARS) as well as an iconic depiction of the nutrient (e.g., the sugar label included a tablespoon of sugar). Similar to the current labeling system implemented in Israel [24], the final warning consisted of a black circle with the text high in added (nutrient) (e.g., HIGH IN ADDED SUGARS), with an iconic depiction in the center (e.g., the sugar label included cubes of sugar). For ease of reporting, hereafter we refer to each warning primarily by its shape (e.g., the octagon, triangle, or circle warning). 

As in previous labeling studies [25], a barcode was selected as the control label because it controls for the effect of putting a label on the front of a package (e.g., size, placement, obscuring branding), while also allowing all participants to respond to questions about a label.

We used the Peruvian labeling guidelines to guide label size and placement [26]. Label size was determined by first sizing the warning labels to approximate specifications of the Peruvian front-of-package labeling guidelines, then sizing other label conditions to roughly match the scale and visual weight of the warning condition. All labels were black in color and were displayed on the top right corner of the products. 

### 2.3. Product Development

Similar to previous labeling studies [27], we selected product categories that are commonly consumed in Colombia and include products which would be high in nutrients of concern (fruit drinks, bread, cookies, and soda) [7]. 

For each category, a graphic designer developed product images that were similar to products and flavors existing in the Colombian market, but were fictional brands in order to avoid the influence of pre-existing consumer preferences for particular brands or products [28]. We used the Pan American Health Organization’s (PAHO) nutritional profile model [21], an international nutritional profile model which has previously been used to study warning labels in Colombia [7], as the framework for determining which nutrient labels the products received. Briefly, products are eligible to receive a warning if they are processed or ultra-processed and exceed thresholds for nutrients of concern (e.g., product contains ≥1 mg sodium per 1 calorie; ≥10% of total energy from free sugars; and ≥10% of total calories from saturated fat). While the PAHO model includes other nutrients (such as total fat, trans fat, and other sweeteners), we focused on sugar, sodium, and saturated fat warnings because these are the ones currently under consideration in Colombia according to local advocacy groups. Thus, the fruit drink and soda received a sugar warning, the cookies received a saturated fat warning, and the bread received a sodium warning. Images of the products can be found in Figure A1 in Appendix A. 

### 2.4. Procedures 

We conducted a between–within subjects experiment in which the between-subjects factor was one of four labels: Control (barcode), octagon warning, triangle warning, and circle warning. The experimental factors were displayed on the front of product packages. The within-subjects factor was the product type on which labels were displayed. 

Participants completed an online survey programmed in Spanish using Qualtrics survey software. After providing informed consent, participants were randomized to view one of the four labels on different products, the first product being a fruit drink. Participants viewed a prompt that read, “The next questions are about drink products.” They then viewed two fruit drinks simultaneously, in randomized order: One fruit drink without a label, and a similar fruit drink with the label to which they had been assigned. They then answered questions about the fruit drink products. Fruit drinks were the product category chosen for the choice experiment because they are commonly consumed in Colombia, and represent a category in which it may be difficult for consumers to identify which product is high in nutrients of concern (in this case, sugar).

Next, participants viewed a series of products. They viewed a prompt that read, “The next questions are about labels on food and drink products. You will look at a few labels on different products and answer questions about each one”. The order of the products (bread, cookies, soda) was randomized within each arm. For each product, the participant viewed a product with their assigned label, and then answered questions about the label and the respective nutrient contained in the label (sugar, sodium, or saturated fat). 

Finally, participants viewed a prompt that read, “The next questions are about different labels”. They were randomized to see one of the three product types without a control label or warning label (bread, cookies, or soda), and below the product, they were asked which label would most discourage them from wanting to consume the product. The response options to the question consisted of the control label, the circle warning, the triangle warning, and the octagon warning. 

### 2.5. Measures 

In the first task, a choice experiment in which participants viewed two fruit drinks, participants were asked to identify which of the two products was unhealthy, which was highest in sugar, and which product they most wanted to buy. 

In the second task, a single product assessment in which participants viewed a series of products with labels, we measured perceived message effectiveness (PME) of the labels, using three items from the UNC perceived message effectiveness scale [29,30] which read “How much does this label…” “make you feel concerned about the health effects of consuming this product?” (range 1–4, from not at all concerned to very concerned); “make consuming this product seem pleasant or unpleasant to you?” (range 1–4, from very unpleasant to very pleasant); and “discourage you from wanting to consume this product?” (range 1–4, from not at all discouraged to very discouraged). 

We also measured whether the label grabbed participants’ attention, led to cognitive elaboration (i.e., thinking about health problems caused by consuming the product), and taught them something new (wording and scale for items can be found in Table A1). For the soda only, we assessed cultural relevance (how acceptable the label would be in Colombian society), liking, ease of understanding, and trust. The latter item was not included in the pre-registration but was included in the analysis due to its utility for ongoing advocacy efforts. 

In addition to label-specific questions, participants were asked to assess questions about the product on which the labels appeared. We assessed participants’ ability to correctly identify whether the product contained excess levels of sodium, sugar, or saturated fat; perceptions of healthfulness; intentions to purchase; and product appeal. 

Finally, the participants viewed all label types, and we asked which of the label options would discourage them most from wanting to consume the product. We also asked whether participants had previously seen any warnings on which the experimental warnings were based (the actual warnings from Chile and Peru, the proposed warning from Colombia’s Ministry of Health, and the triangle-shaped warning). 

We selected PME as the primary outcome because it is sensitive to small differences between label types yet also predicts longer-term behavioral change [31] and has previously been used to evaluate warning messages on products, including sugary drinks [29,30,32]. Secondary outcomes included ability to correctly identify products with excess nutrients of concern, correct identification of unhealthy products, and intentions to purchase products, because these are key steps on the pathway from warning exposure to behavior change [15].

### 2.6. Analyses 

All analyses were conducted in STATA version 16.0. A two-sided critical alpha of 0.05 was used to assess statistical significance. 

We calculated unadjusted means (and standard deviations) and percentages for the primary and secondary outcomes. For our primary outcome, PME, we took the average of the 3 items for each product type (Cronbach’s alpha for each product type > 0.70). We then assessed whether primary and secondary outcomes varied by label type. We used linear regression for continuous outcomes (including PME) and logistic regression for binary outcomes. For outcomes that were assessed using repeated measures for multiple product types, we used mixed models treating the intercept as random at the respondent level to account for repeated measures. These models included the between-subjects factor (i.e., label type), the within-subjects factor (i.e., product type), and their interaction. We used postestimation commands to conduct pairwise comparisons of the predicted means or predicted percentages between each label type. We applied Bonferroni corrections to account for multiple comparisons.

Finally, to assess whether the effect of label type on PME differed by education, we tested for an interaction of warning with education level (specified as low (<high school diploma) vs. high (high school diploma or greater)), and used a Wald chunk test to determine the joint interaction. We used postestimation commands to predict means by label type and education level and conduct pairwise comparisons of the predicted means. 

To descriptively evaluate the most discouraging label, we examined the proportion of participants that selected each label type as the one that most discouraged them from consuming products high in sugar, sodium, or saturated fat. 

Finally, we examined the proportion of participants who had reported seeing the warnings prior to the study. We then conducted a sensitivity analysis of our main outcome, PME, to determine whether results differed when prior exposure to warnings was controlled for. While this analysis was not pre-registered, we determined it was important to ensure there was no potential for confounding by previous exposure to warnings. 

## 3. Results

### 3.1. Descriptive Statistics

Raw means and proportions can be found in Table A2 and Table A3. Sociodemographic characteristics of the sample can be found in Table 1. 

### 3.2. Single Product Assessment of Soda, Bread, and Cookies High in Sugar, Sodium, or Saturated Fat

All warnings elicited higher PME than did the control label (*p* < 0.001 for all comparisons of warning to control), and there were no statistically significant differences between warning types on PME (Table 2). When considering the labels on single products (cookies, bread, and soda), all warnings led to a greater percentage of consumers who correctly identified the product as having high levels of nutrients of concern, from 46% in the control condition to 89% (circle warning), 88% (octagon warning), and 84% (triangle warning) (*p* < 0.001 for all warnings compared to control). The circle warning performed better than the triangle warning on correctly identifying “high-in” products (*p* = 0.039) but was not statistically significantly different from the octagon warning. Compared to the control, all warnings also reduced the likelihood of purchasing products high in nutrients of concern (*p* < 0.001 for all comparisons of warning to control), with no differences by warning type.

Main effects by product type can be found in Table A4. Compared to the other products, soda elicited a higher PME and a higher percent of participants correctly identifying that it was unhealthy, whereas cookies were rated as most likely to be purchased. 

### 3.3. Choice Experiment between a Fruit Drink and Less-Healthy Fruit Drink

When considering the choice between a fruit drink and less-healthy fruit drink, all warnings increased the percentage of consumers who correctly identified which fruit drink was higher in sugar, from 32% in the control group to 83% (circle warning), 84% (octagon warning), and 77% (triangle warning) (*p* < 0.001 for all comparisons of warning to control) (Figure 2; full results reported in Table A5); there were no statistically significant differences between warning types. In addition, all warnings increased the percentage of participants who correctly identified which fruit drink was less healthy, from 41% in the control group to 77% (circle warning), 83% (octagon warning), and 76% (triangle warning)) (*p* < 0.001 for all comparisons of warning to control). There were no differences between circle and octagon warnings, but the octagon warning led to a higher percent of consumers correctly identifying the less healthy fruit drink than did the triangle warning (*p* = 0.042). Finally, all warnings decreased the likelihood of selecting the less healthy product as the one they would most like to buy, from 54% of participants in the control condition to 22% (circle warning), 21% (octagon warning), and 24% (triangle warning) (*p* < 0.01 for all comparisons of warning to control); there were no differences by warning type. 

### 3.4. Interaction of Label Type and Education

We examined the interaction of warning type with educational level on PME (Figure 3). The joint test of interaction of education with warning type was not statistically significant (*p* = 0.412), suggesting that compared to the control, the different warning types perform similarly across education levels.

We also examined the interaction of warning with product type, which was statistically significant for PME (*p* < 0.001). Compared to the control, warnings had the smallest effect on PME for the soda, with trivial difference in PME between warnings (Table A6). However, for both the cookies and the sliced bread, the octagon warning had a larger effect on PME relative to the control than did the circle or triangle warnings. 

### 3.5. Other Outcomes

When shown all labels, 49% of participants selected the octagon warning as the one that most discouraged them from consuming a product high in sugar, sodium, or saturated fat (Figure 4); 21% and 27% selected the circle and triangle warning, respectively, with only 4% choosing the barcode control. 

Other outcomes are reported in Table 3. Compared to the control, all warnings grabbed participants’ attention, led to cognitive elaboration, were easy to understand, and taught them something new (*p* < 0.001 for all comparisons of warning to control). There were few differences by warning type, except that the octagon performed better than the triangle on cognitive elaboration (*p* = 0.010) and better than both the triangle (*p* = 0.007) and circle (*p* = 0.030) on learning something new. Compared to the control, all warnings also led to lower perceptions of healthfulness and lower product appeal; the octagon performed better than the triangle for perceptions of healthfulness (*p* = 0.007). Compared to the control, only the triangle warning was rated as less culturally acceptable (*p* = 0.050). The majority of respondents reported that they liked and trusted the warnings, with the highest proportion of participants reporting that they liked the octagon compared to the control (*p* = 0.001), the triangle (*p* = 0.003), and the circle (*p* = 0.020). Similarly, while all warnings were rated as being trusted compared to the control, the highest proportion of participants reported trusting the octagon warning (*p* = 0.008 for comparison with triangle warning).

### 3.6. Sensitivity Analyses

Finally, we examined participants’ reported previous exposure to warning label types. Overall, previous self-reported exposure was low (<20% of participants for all label types, except that 24% of participants in the circle condition reported they had seen it before) (Table A7). In our sensitivity analyses (Table A8), the main results for PME did not change when previous label exposure was included as a covariate. 

## 4. Discussion

This study assessed participants’ reactions to different types of nutrient warnings, and the impact of these warnings on their perceptions of products high in nutrients of concern (sugar, sodium, or saturated fat). Compared to a barcode control label, all warnings elicited higher PME, meaning that participants rated warnings more highly with regards to making them think about the health harms associated with consuming a product high in nutrients of concern, making the product seem less pleasant, and discouraging them from consuming the product. Compared to the control label, all warnings also helped participants identify products as being less healthy, having higher levels of nutrients of concern, and discouraging them from consuming the products. All warnings also grabbed participants’ attention, were easy to understand, and led participants to think about the health harms of consuming products high in nutrients of concern. These results add to a growing body of research that warning labels are a promising nutrition policy strategy. Moreover, there was no interaction between warning type and educational level, suggesting that all warnings would help promote health equity in Colombia, since they have similar impacts across educational levels.

There were few meaningful differences between warning type with regards to the main outcomes, but the octagon warning showed the most promise overall. The circle warning performed better than the triangle warning with regards to helping consumers correctly identify that a single product had high levels of nutrients of concern, while the octagon warning performed better than the triangle warning with regards to helping consumers correctly identify the less healthy fruit drink. However, the overall pattern of results suggested that the octagon warning modeled after Chile and Peru had a small but consistently greater effect on outcomes relative to the other two warnings, and the triangle warning tended to perform the least well. In addition, the largest proportion of participants (49%) selected the octagon warning as most discouraging. Participants also rated the octagon-shaped warning as being the most liked and trusted. 

One possible reason why the octagon warning elicited higher PME than the other warnings is because the octagon shape may be associated with the concept of caution or danger in the Colombian population. Indeed, the primary difference between the warnings was the shape, although we were not able to explicitly test out the effects of warning shape versus other warning elements in this study. Previous studies of warning shape in other countries have generally indicated that octagons perform similar to or better than other shapes like rectangles or triangles [27,30,33]. Additional qualitative research would be valuable in providing insight on how Colombian consumers perceive octagonal shaped warnings.

Because warning label policies have been implemented elsewhere in the region, it was important to test whether results changed when controlling for prior exposure to different label types. For example, the octagon warning used in this study was based on previously implemented warnings in nearby countries (Chile and Peru) that could appear on imported products in the Colombian market. A similar warning was also used in mass media campaigns promoting labeling initiatives in Colombia [34]. However, surprisingly, few participants reported having previously seen the octagon warnings, and the highest levels of exposure were actually reported for the circle warning. One reason for this may be because the Colombian Ministry of Health had announced its proposal for a front-of-package labeling system, including a warning very similar to the circle warning used in this study, at the end of February 2020 [20]. Regardless, the main results of this study did not change when self-reported previous label exposure was included, suggesting that this was unlikely to influence the observed results. 

We also found that there was a differential effect of warning label by product type, such that warnings had the least impact for soda. This finding is unsurprising, given that Colombian adults may be more likely to already know that soda is high in sugar, and thus the warnings have less capacity to alter consumers’ perceptions about the product or intentions to consume it. Prior research has shown differential impacts of warnings by beverage type, with stronger effects for non-soda beverages [35]. Notably, for both the bread and cookies, the octagon warning elicited a higher PME relative to the control than did the circle or triangle. 

### 4.1. Public Health Implications

These results suggest that a policy requiring warnings on products high in nutrients of concern would help inform the Colombian population and potentially reduce consumption of these products. Such a policy could have meaningful effects, considering recent evidence showing that the majority of packaged foods and beverages in the Colombian food supply are high in sugar, sodium, and saturated fat and would be required to carry a warning according to international nutritional profile models [7]. Although real-world evidence on the impact of nutrient warning labels is nascent, a recent study from Chile found that that after implementation of its labeling and advertising law, households reduced purchases of sugary drinks by 23.7% compared to what would have been expected without the law [18].

### 4.2. Strengths and Limitations

The goal of this research was to compare three different warning label systems which are under consideration by Colombian policymakers, rather than examine the impact of specific design features. Thus, one key limitation of this study is that we were unable to differentiate which aspects of the warnings were influencing participants’ responses. For example, previous research has found that the shape, color, or marker words like “WARNING” can influence reactions to warning messages [30]. Future research will be needed to understand how these different design elements may influence message reactions and behavioral change in the Colombian population. In addition, this study focused on participants’ reactions to warnings, assessments of products, and intentions to purchase products high in nutrients of concern. While these outcomes are part of the pathway from warning exposure to behavioral change, future research may be needed to understand whether warning labels impact consumers’ actual purchasing decisions in Colombia; in addition, it is unclear how warnings will perform against other FOP labeling options that may be considered, such as the Guideline Daily Allowance (GDA) or Nutriscore. Finally, while the online sample was designed to reflect the diversity of the Colombian population, individuals with low literacy, less than a high school diploma, or lack of access to the internet were less likely to be included in this online panel. On the other hand, previous research has suggested that online convenience samples can lead to generalizable findings for experiments [36].

An additional caveat to consider is that this study used the PAHO nutritional profiling system as the basis for applying warnings to products. Like all nutritional profiling systems, the PAHO system attempts to categorize foods, which have many complex nutritional attributes (e.g., processing levels, micro- and macro-nutrients, additives, or others), into a relatively simple, quantitative system which can be readily applied by food manufacturers. The PAHO model, like the Chilean nutritional profile model and others, focuses on identifying products that have high levels of nutrients of concern, particularly as they relate to sugars, sodium, and saturated fats that are added to the product. This type of method may have drawbacks that may be confusing to consumers for some products (e.g., jelly with added sugar may receive the warning label, but jelly with only intrinsic sugars but the same amount of total sugars may not receive the warning label). Because this study only included a small subset of products, future research would be useful to understand how warnings influence consumers’ purchases across an array of product types. 

## 5. Conclusions

Nutrient warning labels on the front of product packaging offer a promising strategy to help Colombian consumers correctly identify that food and drinks high in nutrients of concern are less healthy and discourage them from purchasing these products. The evidence is suggestive that an octagon warning similar to the one used in Chile or Peru would have the most impact on Colombian consumers. More research is needed to understand how warning labels compare to other labeling systems in order to inform an effective policy to reduce overconsumption of food and drinks high in nutrients of concern in Colombia. 

## Figures and Tables

**Figure 1 nutrients-12-03124-f001:**
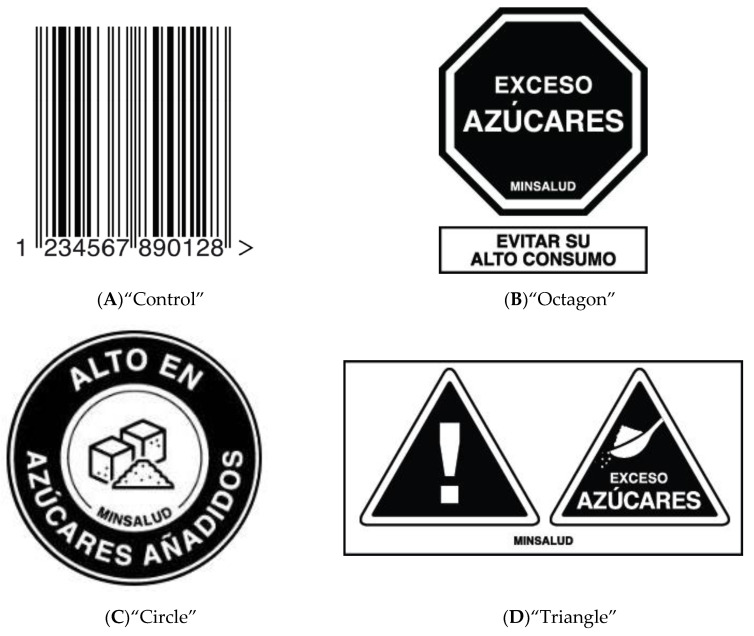
Control label (**A**) and warnings (**B**–**D**) used in experiment. Note. English translations: (**B**) Octagon text reads, “Excess sugars. Ministry of Health. Avoid high consumption.” (**C**) Circle text reads, “High in added sugars. Ministry of Health.” (**D**) Triangle text reads “Excess sugars. Ministry of Health.”.

**Figure 2 nutrients-12-03124-f002:**
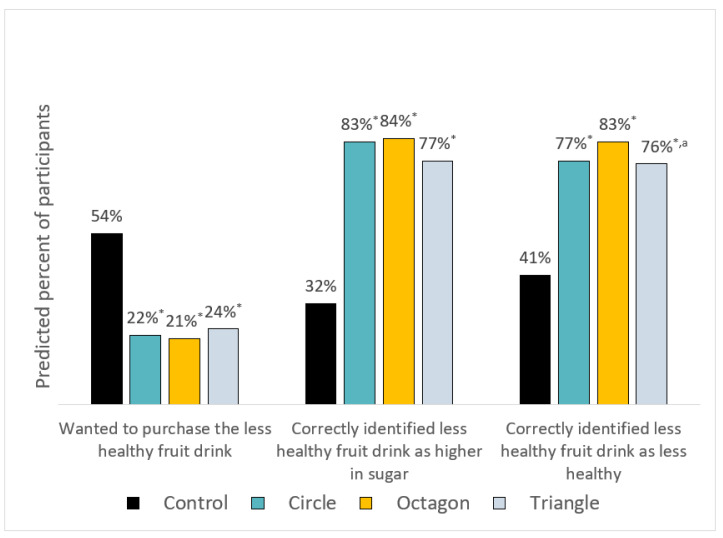
Predicted percent of participants who wanted to purchase the less healthy fruit drink, and percentage who correctly identified the less healthy fruit drink as higher in sugar and as unhealthy by label type. Note. * Warning is statistically different from control, *p* < 0.001. ^a^ Warning is statistically different from octagon warning, *p* < 0.05. There were no differences between circle and triangle warnings. Predicted percent, standard errors, and *p*-values for contrasts can be found in Table A4.

**Figure 3 nutrients-12-03124-f003:**
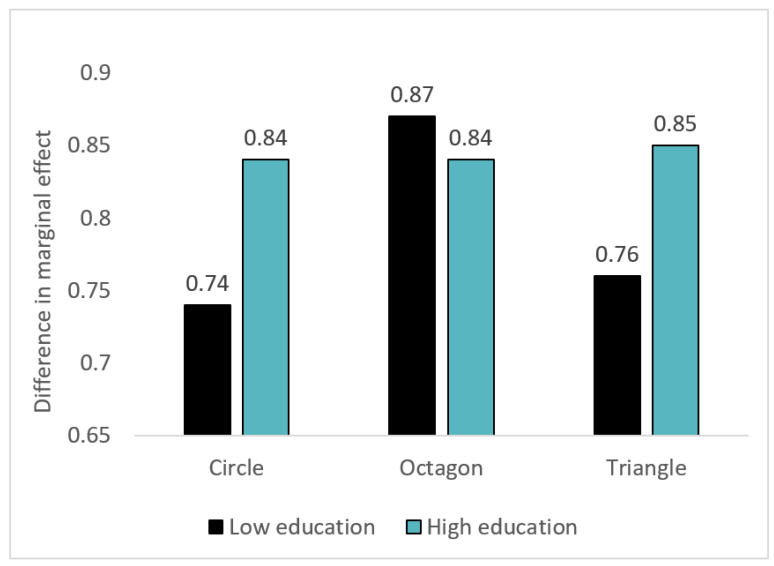
Difference in marginal effect of warnings compared to the control label on perceived message effectiveness (PME) in high versus low education. Note. *p*-value for joint chunk test for the interaction of warning label types and education = 0.412.

**Figure 4 nutrients-12-03124-f004:**
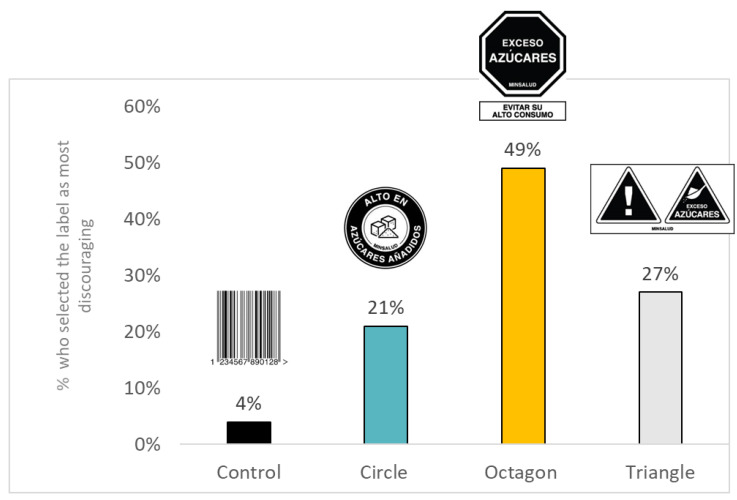
Percent of participants who selected a label as the one that would most discourage them from consuming a product high in sugar, sodium, or saturated fat. Note. The images refer to sugar, but the results are across all products.

**Table 1 nutrients-12-03124-t001:** Socio-demographic characteristics (*n* = 1997).

	*n*	%
Arm		
Control	481	24%
Circle	513	26%
Octagon	503	25%
Triangle	500	25%
Age		
18–24	399	20%
25–34	520	26%
35–44	419	21%
45–54	379	19%
55+	280	14%
Gender		
Male	983	49%
Female	1005	50%
Other gender identity	9	0%
Body mass index (BMI, kg/m^2^)		
Underweight (<18.5)	76	4%
Healthy weight (18.5–24.9)	920	48%
Overweight (25.0–29.9)	676	35%
Obese (>29.9)	253	13%
Education Level		
Low (High school diploma or less)	1001	50%
High (More than a high school diploma)	996	50%
Region		
Atlantica	418	21%
Oriental	395	20%
Central	491	25%
Pacifica	330	17%
Orinoquia	21	1%
Bogota	342	17%
Children		
Yes	1309	66%
Ethnicity		
Indigenous	58	3%
Black, mixed, afro-descendent	242	12%
Other ethnic group	390	20%
No ethnic group	1307	65%

Note. Characteristics did not differ by between-subjects experimental arm. Missing demographic data ranged from 0% to 3.61%.

**Table 2 nutrients-12-03124-t002:** Overall predicted means and percentages by label type, for perceived message effectiveness of the label and assessment of a single product high in sugar, saturated fat, or sodium.

	PME			Correctly Identified Product as Having Excess of a Nutrient			Likelihood of Purchasing the Product if It Were Available		
	Mean	SE	*p*	%	SE	*p*	Mean	SE	*p*
Warning									
Control	1.79	0.02	n/a	46%	1.9%	n/a	2.59	0.03	n/a
Circle	2.59	0.03	<0.001	89%	1.1%	<0.001	2.01	0.03	<0.001
Octagon	2.65	0.03	<0.001	88%	1.1%	<0.001	1.97	0.03	<0.001
Triangle	2.61	0.03	<0.001	84% ^a^	1.3%	<0.001	1.99	0.03	<0.001

Note. SE = standard error; n/a= not applicable; PME = perceived message effectiveness. *p*-value is for the contrast between each warning label and the control. ^a^ Warning is statistically different from circle warning, *p* < 0.05. There were no statistically significant differences between the triangle and octagon warnings.

**Table 3 nutrients-12-03124-t003:** Predicted means and percentages of reactions to label types and perceptions of products.

	Control			Circle			Octagon				Triangle	
	Mean	SE	*p*	Mean	SE	*p*	Mean	SE	*p*	Mean	SE	*p*
Grabbed attention	2.23	0.03	n/a	2.49	0.03	<0.001	2.57	0.03	<0.001	2.51	0.03	<0.001
Made them think about health problems from consuming the product	1.79	0.03	n/a	2.74	0.03	<0.001	2.84 ^a^	0.03	<0.001	2.68	0.04	<0.001
Would be acceptable in Colombian society	2.70	0.03	n/a	2.60	0.04	0.236	2.65	0.04	1.000	2.57	0.04	0.050
Would be healthy for a child aged 1 to 12 to consume the product every day	2.28	0.03	n/a	1.60	0.03	<0.001	1.56 ^a^	0.03	<0.001	1.69	0.03	<0.001
Perceived product appeal	2.68	0.03	n/a	2.16	0.03	<0.001	2.16	0.03	<0.001	2.14	0.03	<0.001
	**%**	**SE**	***p***	**%**	**SE**	***p***	**%**	**SE**	***p***	**%**	**SE**	***p***
Liked the label	60%	2.2%	n/a	63%	2.1%	1.000	71% ^b^	2.0%	0.001	61%	2.2%	1.000
Easy to understand the label	68%	2.1%	n/a	92%	1.2%	<0.001	93%	1.2%	<0.001	89%	1.4%	<0.001
Label taught something new	30%	0.7%	n/a	75%	0.7%	<0.001	77% ^b^	1.2%	<0.001	74%	0.6%	<0.001
Trusted the label	49%	2.3%	n/a	67%	2.1%	<0.001	73% ^a^	2.0%	<0.001	63%	2.2%	<0.001

Note. SE = standard error, n/a= not applicable. *p*-value is for the statistical significance of the difference between each warning label and the control. ^a^ Warning is statistically different from triangle warning (*p* < 0.05). ^b^ Warning is statistically different from triangle warning (*p* < 0.05) and circle warning (*p* < 0.05). No other statistically significant differences between circle, octagon, and triangle warnings (*p* > 0.05).

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
