# Peer review of "Designing an Effective Front-of-Package Warning Label for Food and Drinks High in Added Sugar, Sodium, or Saturated Fat in Colombia: An Online Experiment"

_nutrients, 2020, doi:10.3390/nu12103124_

Round 1
Reviewer 1 Report
I think this is a good paper and would be improved by doing a few things.
1. I STRONGLY suggest removing mention of ultra-processed foods (UPFs) as it does not advance or apply to your argument about sugar. In the Monteiro NOVA classification, UPFs are defined as any manufactured food with more than 5 ingredients and include a wide array of foods that have sugar and do not have sugar e.g. from sugar-sweetened beverages, to bread sold in packages to infant formula, foods for special medical use, and breakfast cereals that have added nutrients regardless of their sugar content or nutrient contribution.
2. I suggest that the authors state what the sugar level was on the products that would be deemed as high on their label 3. I would have a warning about extrapolation of these data to all foods with sugar, especially foods that provide nutrients of concern due to their underconsumption. The label works well on so-called fruit drinks, most of which have low nutritional value. This research shows that a warning label system appears to promote a proper nutritional choices. My fear comes from extrapolation of these data for use on nutritious products that contain added sugar creating a scenario where the consumer would avoid a food, that when used properly, is helpful to or needed in the diet. Two examples come to mind. First, consider dried cranberries with important antioxidants and other components. When eaten without added sugar, most reject them for being too tart. (Sugar is added making their sugar content comparable to that found naturally in raisins.) The warning label might discourage the eating of the cranberries when in fact it is not less healthy than choosing dried cherries or raisins. Another example is bran cereal. Miller's bran is a fabulous source of fiber. However, in its unaltered form it is acceptable to horses but not humans because of its cardboardy taste and texture. With added sweeteners, processing and other ingredients , various ready-to-eat bran cereals provide 6-11 g of dietary fiber per serving, as well as many other nutrients. Consumers throughout the world fail to meet the fiber requirement and ingest about half the recommended level. Thus warning labels makes bran cereal ( despite its added sugar), a food not associated with weight gain ( in fact it is associated with weight maintenance). More importantly is provides a substantial amount of an underconsumed nutrient dietary fiber. Application of such a warning sign to foods that help the diet might discourage the choice of foods that should be included in the diet. This must be talked about in the discussion.
3. If such a labeling system is implemented, there must be care so that some of the mistakes such as those that occurred in Chile do not occur. Butter in Chile is an example. Unsalted butter contains no warning stop sign, but salted butter contains 2 warning stop signs - one for salt and one for saturated fat. This is a scenario where the warning signs confuse the consumer and would make them think that unsold butter has not saturated fat. This could happen with sugar where a jam is made with added sugar - it would contain a warning sign, but a jam made by boiling fruit down to the jellying point would have the same sugar content in the diet but would not have the label so the consumer might believe that one is healthier when in fact both choices should be eaten sparingly. Such things need to be addressed in the discussion of the ramifications of the use of such labels.
4. Labeling of breads with a sugar warning may be misleading. Labeling laws require that the ingredients in order by weight of amount added must be declared on the ingredient statement, but in breads containing a small amount of added sugar (not a sweet dough) some, if not all, of the sugar is consumed by the yeast during fermentation to produce carbon dioxide that leavens and alcohol and other flavor (Aldehydes and ketones) compounds that characterize yeast bread's alluring smell and flavor. So when the bread is eaten, the amount of sugar on the label is more than the amount being consumed.
Reviewer 2 Report
This is a very well-written manuscript and covers a very important issue regarding consumer attitudes towards warning labels of energy-dense items. I have some minor comments for the authors:
-The methods do not make it clear whether the same food products (in Appendix A) were used to show the different types of warning labels. If not, please explain.
-For the choice experiment, please explain why fruit juice was used instead of other food items.
-Line 187: please do not start sentences with abbreviations. Please check throughout the manuscript.
-While the authors mention it briefly in their limitations, I think a big factor that would influence participants' beliefs and the results would be the message content and how the message was framed etc. For example, the message on the circle is not the same as on the octagon and triangular warnings. How does that affect the accuracy of results being presented here?
-Were the warning labels placed at the same spot on the food items? If not, that would be a major limitation.
